# The role of informal support systems during illness: A qualitative study of solo self-employed workers in Ontario, Canada

Tauhid Hossain Khan[1]*, Ellen MacEachen[2]

1 Department of Sociology, Jagannath University, Dhaka, Bangladesh, 2 School of Public Health Sciences, University of Waterloo, Waterloo, Ontario, Canada

* th3khan@uwaterloo.ca

**Data Availability Statement:** There are ethical restrictions which prevent the public sharing of minimal data for this study. Data are available upon request from the Research Ethics Board at University of Waterloo via phone (1-519-888-4567

## Abstract

Today's labor market has changed over time, shifting from mostly full-time, secure, and standard employment relationships to mostly entrepreneurial and precarious working arrangements. In this context, self-employment (SE), a prominent type of precarious work, has been growing rapidly due to globalization, automation, technological advances, and the rise of the 'gig' economy, among other factors. Employment precarity profoundly impacts workers' health and well-being by undermining the comprehensiveness of social security systems, including occupational health and safety systems. This study examined how self-employed (SE'd) workers sought out support from informal support systems following illness, injury, and income reduction or loss. Based on in-depth interviews with 24 solo SE'd people in Ontario, Canada, narrative analysis was conducted of participants' experiences with available informal supports following illness or injury. We identified three main ways that SE'd workers managed to sustain their businesses during periods of need: (i) by relying on savings; (ii) accessing loans and financial support through social networks, and (iii) receiving emotional and practical support. We conclude that SE'd workers managed to survive despite social security system coverage gaps by drawing on informal support systems.

## Background

Today's labor market is constantly changing; self-employment (SE) has grown as a key non-standard, precarious, and contingent work arrangement internationally [1–4]. SE is part of a "paradigm shift" from managerial/manufacturing capitalism to entrepreneurial capitalism in the present digital era, appearing in different forms than it did 50 years ago [5,6]. The proportion of the precarious work, including SE, has been growing rapidly in recent decades due to globalization, automation, and the information revolution [1,3,7,8]. It has been estimated that non-standard employment accounts for more than 60% of workers worldwide [9,10]. For instance, in Canada in 2019, 2.9 million people, or more than twice as many as in 1976, identified as self-employed (SE) [11]. In fact, SE'd workers represent 15% of employment in Canada [11]. Similarly, 10% of the Australian workforce was SE'd in 2016 [12], and in 2017 SE'd

ext. 36005) or email (oreceo@uwaterloo.ca) for researchers who meet the criteria for access to confidential data.

**Funding:** This research was funded by the SSHRC (Social Sciences and Humanities Research Council)CIHR (Canadian Institutes of Health Research) Productive Workforce Partnership Grant (# 895-2018-4009 and #159064). EM: received the grants websites: https://cihr-irsc.gc.ca/e/193.html https://www.sshrc-crsh.gc.ca/home-accueil-eng. aspx The funders had no role in study design, data collection and analysis, decision to publish, or preparation of the manuscript.

**Competing interests:** The authors have declared that no competing interests exist.

workers comprised 15% of the workforce in Europe [4]. Because of the growing 'gig' economy and the breakdown of traditional employment institutions that provided secure, lifetime positions with predictable promotion and stable income, this SE trend is intensifying [3,13–15]. Against this backdrop, scholars have noted that the existing social security systems for workers need to adapt to the new labor market [3].

SE'd workers have been portrayed as a special group of homogenous people in the research literature [8], implying that they possess good health, enjoy the freedom of being their own boss and flexible working hours [4,16], do not rely on the states (e.g., social security protection), and enjoy greater job satisfaction, quality of life, and opportunity to gain work-life balance than employees [2,4,16,17]. They have a reputation for taking on a substantial amount of personal risk in order to build their enterprises and also create job opportunities for others [4,8,13,18]. However, mounting international evidence stresses that the changing nature of work is having profound adverse effects on workers' safety, health, and wellbeing [4,7,10,18–21]. The entrepreneurial depictions, mentioned above, do not reflect the recent reality of the SE, where a significant number of SE'd workers in a given society are compelled to undertake this type of work due to unemployment and scarcity of alternatives [4,18,22–27]. For example, SE'd workers in some sectors are at higher risk for physical and mental health hazards such as musculoskeletal disorders, joint pain, sleep disorders, and digestive complaints, compared to salaried workers [18,28,29]. These risks stem from the nature of SE work, for example, some SE'd people encounter a higher level of job demands and workloads (e.g., farmers), self-exploitation (drudgery), isolation due to working alone, reputational threat, customer and contractor betrayal, absence of social protections (e.g., lack of health insurance), and elevated anxiety about financial matters due to volatile income [18]. In addition, scholars have claimed that the dominant narrative that SE'd are healthier than salaried workers [4,10,18,21,29–31] can be explained by the 'selection effect' [18]. The diversity of SE'd workers is described by the Law Commission of Ontario (2012), which noted that: "the experiences and vulnerabilities of this group range from billionaire entrepreneurs to taxi drivers working 90 hours a week simply to pay their bills and includes many people who are gaining income from self-employment activity alongside their main job" (LCO, 2012: 75). Therefore, SE does not always mean self-sufficiency. Instead, some SE'd workers are considered precarious workers at risk of poverty and social exclusion because they have low job and income security, poor working conditions, and limited social safety net coverage [3,4,10,21,32]. In all, SE appears to be a "a double-edged sword" [18].

Across jurisdictions, SE'd workers are largely excluded from many social security protections, including the workers' compensation coverage, employment insurance, and state pension plans [3,7,10,21,33]. The ILO's study of G20 countries found a social protection coverage gap for SE'd workers in many countries [34]. In some countries (e.g., Estonia, Latvia, Portugal, and Slovick Republic), 40–50% of precarious workers were less likely to receive any form of income support when they were out of work due to injury, sickness or any form of impairment [10,14,21]. In this context, Australia (NSW) and Canada (Ontario) are similar, providing limited and partial support for the SE'd workers; with Australian SE'd workers having more schemes to opt into than Canadian workers [21,33,35]. However, some welfare states play pivotal roles in terms of protecting SE'd workers [10,21]. For example, Finland provides a broad support system to workers regardless of employment status, in which SE'd workers are covered with earnings-related pension schemes (old-age pension, disability pension, survivors' pension) and have access to a universal basic social security system (parental and sickness benefits, housing and unemployment benefits) [21,36]. In all, the absence of a social safety net can perpetuate the distress of SE'd workers; mounting evidence illustrates a strong relationship between precarious employment and poorer health outcome [4,10,21,37] and numerous social costs [18,21,28].

Statuary social supports for SE'd workers world-wide are partial and scanty [10,21]. The aim of this study was to examine how, in the absence of such supports, SE'd workers fared when they were injured or ill and unable to work. The focus of this paper is on how SE'd workers turned to informal support systems following illness, injury, and income reduction. By informal support systems, we refer to the SE'd person's system of primary relationships with individuals such as family members, friends/social networks, and relatives or neighbors [38], including instrumental, emotional, and informational supports [16]. Hilbrecht (2016) described instrumental supports as including practical assistance concerning concrete skills, actions, or resources, for example, financial support from family members. Emotional supports include providing empathy, reassurance, and understanding; for example, providing counseling and showing empathy towards SE'd workers during economic volatility. Finally, informational supports offer information or suggestions, such as reliable guidelines and information about government-provided support/benefits [16].

## Methodology

### Study design

A qualitative methodological approach was utilized for this study due to our interest in how SE'd workers' fared when experiencing ill-health and inability to work. In line with this approach, an interpretative paradigm, which focuses on the understanding of phenomena through meanings people bring to them, was used to reflect upon the narratives provided by participant [39]. This approach helped to unpack the underlying meanings embedded in SE'd workers' stories, including everyday practices and experiences situated in a larger cultural context. The study was approved by the Research Ethics Board of the University of Waterloo, Canada.

### Participants, sampling, and recruitment

Participants were selected for this study based on the following inclusion criteria: solo SE'd workers (i.e., no employees), aged 18 years or older, experience of illness or injury while SE'd (work-related or not), main income is from self-employment, and fluent in English (due to researcher language limitations) (Table 1).

Various social media platforms were used to recruit participants from Ontario, Canada, including, Linkedin, Facebook, Kijiji, Twitter, and Tumblr. From among eligible participants, we selected participants purposively for information-rich and heterogeneous cases (Patton, 2001). The 24 participants included in the study were between 21 and 62 years of age, with varied education (college diplomas, university degrees, etc.) and income levels ($25k/year—$200k/year). Similar proportions of men and women were included in the study. The workers were interviewed by the lead author using audio/video conferencing with Zoom and WhatsApp. The interviews were conducted between January and July 2021 and lasted 1.10 hours on average.

### Data collection

As this study involved soliciting solo SE'd workers' personal experiences including culturally sensitive information (e.g., income, sickness, personal family lives), a semi-structured, in-depth interview approach was selected to give time and space to each person to explain their situation. Interview questions were informed by literature and discussion with the research team. We used a combination of questions and probes (follow-up questions) to achieve breadth of coverage across the following key topics: (a) work-related experiences; (b) illness,

**Table 1. Participant characteristics.**

| Pseudonym | Gender | Age | Education | Type of SE'd work | Type of illness/injury | F. Income (CAD)/Year |
|---|---|---|---|---|---|---|
| 1.Habibur | M | 22 | College diploma | Uber Driver | Depression Leg fracture | 50K |
| 2.Tasmina | F | 32 | College diploma | Home childcare | Flu/ fever | 50K |
| 3.Emma | F | 36 | Undergraduate degree | Catering | Pneumonia | 25K-50K |
| 4.Mamun | M | 45 | Graduate degree | IT consultant | Spinal injury | 45K |
| 5.Zayan | M | 22 | College diploma | Gig food delivery: Door Dash Skip the Dishes | Broken ankle | 100K |
| 6.Ruby | F | 42–47 | Graduate degree | Rotary Public commissioner | Depression stress, obesity | 25K-50K |
| 7.Patrick | M | 62 | Undergraduate degree | Actor, catering | Knee injury | 50K-100K |
| 8.Sarah | F | 54 | Graduate degree | Property manager | Stomach pain | 50K-100K |
| 9.Sumon | M | 22 | College diploma | Food delivery | Broken hand | 25K-50K |
| 10.Mary | F | 46 | High school | Fashion design | Sjogren syndrome | < 25K |
| 11.Faria | F | 21 | Undergraduate degree | Beautician | ADHD | 25K-50K |
| 12.Remi | F | 45 | College diploma | Financial advisor | Asthma, Covid-19 | 50K-10K |
| 13.Sarika | F | 50 | High school | Cleaner | Sleep disorder | 25K-50K |
| 14.Scott | M | 50 | College diploma | Construction | Arthritis | 50K-100K |
| 15.Ander | M | 25 | Postgraduate diploma | Online business/ E-commerce | Anxiety, stress, depression | 25K-50K |
| 16. Bob | M | 33 | College diploma | Singer, DJ | Anxiety, stress back pain | 25K-50K |
| 17.Jane | F | 33 | Undergraduate degree | Actor, Writer | Nervous system disorder | 130K |
| 18.Jimmy | M | 35 | Graduate degree | Data analyst | Regular migraine headaches | 200K |
| 19. Paul | M | 32 | College diploma | Electrician | Backbone Injury | 50K |
| 20. Ayla | F | 35 | College diploma | Grocery business | Cardiology ADHD | 50K-100K |
| 21.Miller | M | 24 | Undergraduate degree | Music trainer, musician | Leg injury | 50K |
| 22.Mila | F | 35 | Graduate degree | Tailoring | Backpain, fatigue | 50K-100K |
| 23.Arnob | M | 30 | Graduate degree | Debate /public speaking trainer | Anxiety, stress, burn injury, depression, | 25K-50K |
| 24.Pablo | F | 26 | College diploma | Financial advisor | Stress | 25K-50K |

injury or income reduction/loss; (c) government and informal social benefit systems they used; (d) health and wellbeing in the context of work. Interviews were audio-recorded and transcribed verbatim by professional transcriptionists. Oral informed consent was received before interviews started, documented through the transcriptions, stored in a safe location, and approved by the ethics review committee. Along with reflexive journal, detailed field notes were taken after each interview to describe encounters, including the immediate impressions and context, and analytic insights.

## Data analysis: Thematic narrative analytical approach

Following Reissman's (2008) Narrative Thematic Analytical Approach (NTAA), this study aimed to gain insight into the experiences and practices of SE'd workers as told stories (narratives) pertinent to their life experiences [40,41]. NTAA is well fitted for the context of this study because, unlike other types of narrative analysis, it focuses on "what content a narrative communicates [what is told or spoken], rather than precisely how a narrative is structured to

make points" [42] [p.81]. The analysis was composed of several phases: reviewing the transcripts multiple times, developing a codebook, establishing themes and subthemes, and identifying core narrative elements associated with each theme. A combination of both deductive and inductive coding was used during the data analysis process, resulting in a codebook of 10 codes. The codes were informed by the existing literature, and issues identified during interviews. Using Nvivo, the data sets were re-arranged in terms of the codebook. These codes helped us reflect on the overall patterns of the data, including identifying common themes. Our analysis resulted in the development of three major themes, as discussed below.

## Findings

The findings describe participants' stories about the navigation of the informal support systems, following their illness, injury, and income reduction and/or loss. This section begins by describing how participants managed by drawing on their own savings, and also how low-wage SE'd workers, such as gig workers, were unable to create a savings pool. The next section describes stories about alternate forms of financial support that SE'd workers relied on when they were ill or injured, such as loans from family. Finally, we describe emotional and practical support as the third main type of informal support that sustained SE'd workers in times of need.

### Relying on savings

SE'd workers in this study placed an emphasis on the need to have personal savings in order to get through income fluctuations. As noted by one participant: "I used my savings to back up everything" (Habibur). The participants voiced the idea that SE'd workers are the architect of their own fortune and wellbeing. As SE'd workers were not required to contribute to salary replacement insurance or benefits, they believed that neither the government nor private organizations were obliged to protect them. As one SE'd worker reflected: "There is no help as a self-employed person, I cannot claim anything until I become absolutely disabled" (Remi). The SE'd participants took for granted that they had to save money for future for things like medications, vacations, and pandemic. Participants described relying on savings during periods of illness, as per the following instance: "Well, at that time there was no support systems by the government [. . .] My support system is my own saving money . . . So basically, when I was sick, I was solely on my own money that I could survive" (Remi). Similarly, a beautician (Farina), noted that SE'd people are obliged to save money given that they have little access to formal supports. When prompted to reflect on their access to social security system supports, SE'd workers in this study felt that, because they pay taxes to the government, it was unfair for them not to have access to income replacement protections following illness or injury or, in the circumstances of the income reduction or loss. In this regard, one SE'd worker noted, "I think everyone has the right to have housing and food and you do not worry about those things" (Jane).

A challenge with relying on savings is that low-income SE'd workers, such as gig workers who constitute a growing portion of the SE'd population, are unlikely to be able to set aside savings and thus may have no funds to support themselves during an illness or injury. For example, as Canadian healthcare coverage does not cover the cost of medication, low-wage SE'd participants in this study struggled with the financial challenge of accessing medication and non-core health care:

I have to pay for my meditation from my own pocket. I have to pay for my IV therapy, a small fee, because part of it is covered under OHIP [Ontario's free public health insurance].

I have to pay for the transportation to my appointments, my regular doctors' appointments. . . .. My neurologist. . . .. I haven't had to see them for a while with COVID. . . .. Blood work is covered . . ., except what I need a special test every now and then. And it's $60. Like an hour's massages [is] $80 to a $120. I don't have money for that. So, I bought a massage pad to try to help ease those symptoms. . . . I also look for opportunities like . . . college students for like, massage therapy, osteopath, those kinds of things where they need people to practice on and they'll do it for free. So, I look for things like that . . .where I can get at least treatment . . . at no cost to myself. Because I just don't have any extra money. (Mary)

In some cases, SE'd workers managed their lack of funds by not buying required medications: "I . . . don't take medication either for it. So, because we can't afford it [. . .] Well, health care is free in Ontario, but medications I can't afford them". Describing scarce income, a SE participant similarly said, "I would rather spend this much money on groceries rather than on medicine. However, medicine is important" (Ander). Drawing on savings was easier for the higher-income SE'd in this study. For example, a SE'd financial advisor was able to rely on her savings when ill and additionally had been paying into private critical illness insurance:

I have no choice. I have to use my own savings [. . .] Yes, yes, I have saved so much that I can do that. I have enough money to support myself for like two or three years. [. . .] I have insurance, for . . . like, critical illness. If I become critically ill, I have disability insurance, I have savings and that's it. So, I just survived. So yeah, my earning [. . .] and my passive income . . . and there is no help from the Government. I'm self-employed so, everything I pay for myself, my own disability or critical illness. [. . .] Only the plan that I purchase my-self through insurance company for critical illness like cancer, heart attack, stroke, other illnesses and then my disability, if I become disable [. . .] I wouldn't trust the Government. No, no, no, I wouldn't trust the Government . . . I'm paid into EI many years, jobs before I'm paid into one time I had to claim, and they ask for back, I don't trust the Government. (Remi)

## Loans and financial support through social networks

Credit card loans were another way that SE'd workers managed financially when their income was low. For some, this meant a cycle of loans and interest, and more payments, as per this SE'd participant: "My savings was very poor . . . [I had] not enough to support my unworked period of time. So, I had to charge my credit card a lot. And after work I have to pay those" (Mamun). All of the SE'd workers in our study described social networks (e.g., siblings, parents, family members, marriage partners, former spouses) as essential support systems during times of need due to illness or injury. These networks provided varied types of support, including financial, reciprocal (e.g., babysitting for a neighbor), mental, and emotional support.

Most of the SE'd workers borrowed money from either family members or friends to stay afloat during difficult times, such as when they become ill. For instance, when a gig worker, who was also an international student, had to stop working for three months due to an injury, he was without an income and was financially supported by his parents. It was incredibly demanding to bear living and medication costs for long in tandem with international tuition fees, as these students usually fund their own living expenses by working, with tuition paid by their family:

"At that time, I got some money from my parents, like they are in Bangladesh right now. So, they help me to go through it for two months. So yeah, I need a few supports so they

offer me something, but I felt pressure to start my work, and returned to work, even though I was not ok . . . I mean I was sort of forced to return, I was afraid to ask [my parents for] more money for the next month" (Sumon).

Most participants discussed both family and friends helping, in combination. For instance, a SE'd electrician described how his family was why he did not worry about support during times of need: "Yes. I had a very good. . . family [. . .] and friends that I've known for all my life. So, I'm not worried [about support if needed during my illness or income loss]" (Paul). In another case, a SE'd cleaner whose earnings were at subsistence level relied financially on both her husband and her mother when she could not work and earn money following illness or other reasons: "If I do need financial assistance [initially she usually asks her husband]. If it was over three months, I would have to rely on my mom" (Sarika). As well, a SE'd rotary public commissioner described approaching her family members for financial support and then sometimes approaching friends: "My parents are big support system right now for me. So, if I desperately need money, my dad gives me a loan. Usually sometimes he can do it, sometimes he can't [. . .] Before my friends, my extended family, my brother, my sister".

Financial support was also described as reciprocal. For instance, a SE'd tailor noted that when she was short of income, she asked friends to help pay her rent, and she did the same for them: "Sometimes we had to face money shortage during my illness or failing to deliver order on time, then I borrow money for house rent from friends, as I have few friends here" (Mila). Likewise, a SE'd licensed home-based childcare provider (Tasmina) described being supported by her husband when her income was low, and also supporting her husband financially when needed.

Some SE'd workers also described members of their community as providing financial support when they had to leave work or had reduced working hours due to being ill with the Covid-19 virus. This support came in the form of donations of money, food or loans with zero interest and flexible time for repayment. For example, one gig worker and international student received donations from his ethnic community when he was ill and unable to work: ". . .there are many other brothers who used to help me in my impoverished times. Even, when I have issues with my tuition fees, I get help from them" (Habibur). He further added: "[. . .] who used to give me a lot of financial help through the time I have problem. Or my sickness or even though I have problem with my tuition fees. I always just . . . call him on say something I have a problem. Whether I have to say just send me the money and I will just pay him. And it doesn't take any . . .. He is helping not only me but other international students".

## Emotional and practical support

Emotional, mental, and motivational support also helped SE'd workers in this study during their times of need and were provided by friends and family. For instance, a SE'd property manager expressed gratitude for the emotional supports to her friends: "Uh just emotional, no one's giving me money. But emotional support or informational support. You know, they'll tell me, recommend what I should do". Similarly, a SE'd fashion designer emphasized her friends' mental and emotional support:

"I have incredible friends. I have a lot of really great emotional support and I used to play Roller Derby before I got sick. There's a lot of fun and I made lifelong friends there. My best friend is incredible. She lives in Vancouver, but we talk every day [. . .] I have great friends . . . they're amazing and they will come and visit me, they will call me". (Mary)

Other SE'd workers in this study (Tasmina, Patrick, Sarika, Habibur, Bob) described how their friends supported and helped them mentally and emotionally by providing food, lending a hand to help with work, and taking care of children. In this context, one SE'd participant who ran a DJ business (Bob), described a situation when he was committed to perform as a DJ in a wedding program, but could not make it due to a sudden illness. He described how his friends performed on his behalf, thereby saving his professional reputation. Although he paid his friends for this support, they nonetheless helped him at a difficult time:

> If I get injured, say my back or something like that, I'm still able to do the job but I might need help carrying the heavy equipment. In that case, I have a few friends . . . For bigger jobs where I need same extra vehicle, or I have a lot of setups . . . in those cases I just call one of those people that I can trust that knows the job and they come out and help me. They will do all the heavy lifting . . . and I pay them as well. So. . . that comes out of my pocket. They are not employees or anything like that but I also feel like, you know, some friends will just help you out. . . . So I always make sure do that and they are always ready to help me again, sort of the thing right?

Roommates were also described as helping the SE'd workers following their injury or illness: "[My roommates] help me to cook, they help me to get go to the doctor, buy some medications, that's all. They help me to buy stuff like I need to buy some goods, they will buy for me." (Sumon)

The SE'd workers also described their family as an important mental health support system (in addition to the financial support provided). For instance, one SE'd worker was supported by both their mother and husband:

> Oh, I don't currently get financial support from anybody. But . . . I do in terms of emotional support. . . .My mom and my partner, all are very supportive, they have all seen me struggle with my health issues over the years. So yeah, I do have a great support system that makes a big difference. (Sarika)

In another example, in another case when a SE'd worker was ill and without income, her husband stepped in to provide childcare that she could no longer afford to pay for: "Ah, my ex-husband has been really great. When I got sick, the kids were a little younger and he took over as primary parent . . . they went to live with him full time" (Mary). Likewise, spouses were described as supporting their SE'd partners mentally and physically following an illness or injury. For instance, following an injury, a SE'd IT consultant's wife took a leave from her work to help him. As he explained: "When I got injured that time my wife had to take also leave for some days [. . .] and that's why her work was also impacted" (Mamun). Similarly, the wife of a SE'd data analyst (Jimmy) managed his client's email and dealt with the issues that were not too technical when he was ill.

Other close relatives also helped the SE'd workers when they were ill or injured and unable to work. For example, a SE'd home childcare provider how her in-laws helped her during her illness by providing food and care for her children:

> As my family doesn't live over here [in Canada], and my in-laws live with me, [when I get sick] so they can cook. If I cannot cook for my sickness time . . . they help me—like if I need support to take care of my children because if I'm sick. (Tasmina)

To sum up, participants in our study described getting through illness, injury, job loss or income reduction by relying on savings to support themselves and their families. They also relied on their family, friends, and other social networks for loans, and relied on emotional, mental, and pragmatic supports.

## Discussion

While employees in regular employment relationships are protected by statuary and employer support systems, most SE'd workers, including those we studied in Ontario, were not fully covered in this context. The limited social security support for SE'd workers is well addressed in a mounting international literature [3,7,21,33,43–45]. However, very little research has addressed how SE'd workers' survive during times of injury or illness by drawing on informal support systems, with the exception of Hilbrecht [16]. It is historically evident that informal support systems, such as social network, family relationship, kinships, and friendships play a pivotal role in times of distress [46,47]. Against the backdrop of historical role of such relationships and social security system coverage gaps, we discerned how this relationship supported SE'd workers, following their illness, income reduction/job loss. In this context, previous researchers have examined the nexus between family and work-life balance and conflict among SE'd workers [48,49], the effectiveness of informational (e.g. advice) and instrumental (e.g., financial aid) social supports for SE'd people [50–52], whether more support (informational and instrumental) leads to better outcomes for SE'd workers [52], gender relations of SE'd workers in respect to work-family conflict [53], and how family and community as a resource of supports play out for SE'd men and women differently [54]. These studies mainly focused on how work-family relations are positively or negatively affected by their SE'd work or vice versa. In short, previous research has shed light on SE'd workers' experiences with receiving emotional and instrumental supports, and also focused on a work-family context [16].

We found that SE'd workers in our study used informal support systems to fill gaps in social security protection. This support took the shape of personal savings, loans from family and personal networks, and emotional and practical support from family and friends. Altogether, these supports can be categorized into cultural and social capital [55], which are widely described as influential impetus of social development, career growth, and success in self-employment [47,52,56–58]. Moreover, these informal support systems maintain people's happiness and wellness ("good life"), which is historically addressed in classical scholarship [59–61]. Although we were unable to find any previous studies showing the nexus between family/social networks and the social security of SE'd people, studies have documented that entrepreneurial behavior and success are significantly influenced by their family and kinships [62]. These kinship support systems provide benefits through learning (e.g., sharing failure or success experiences) and complementarities, risk-sharing, and lower transaction costs [62]. Our previous findings based on this data set suggested that that some SE'd workers did not trust social security systems to support them in times of need [10,33]. In turn, this may foster relatively more reliance and trust by the SE'd on informal support systems, such as family and social networks.

In our study, family members of SE'd were their key source for a wide range of informal support. Kim, Longest and Aldrich [52] argues that these relationships are instrumental to launch SE'd businesses and for them to thrive and survive. Our findings, however, differ with Hilbrecht [16], who found that SE'd workers had conflicted feelings about financial support from spouses. In our study, SE'd workers welcomed the emotional, mental, financial, and practical supports provided by family and social networks. Welfare theorists and social scientists have been castigated for ignoring the role of family in people's welfare [46]. Akram and

Maitrot (2022) stressed that the family is the pillar of all welfare systems; however, their analyses were built upon the lower middle-income countries and cannot be expected to apply to advanced economies that emphasize individualism [46]. However, our study suggests that, for SE'd workers in high income countries, families maintain an important welfare role.

SE'd workers in this study described non-family networks as also important sources of support to them when injured or ill. These supports included providing food during illness, lending a hand at work, taking care of children, and also financial support. This help was sometimes reciprocal and sometimes paid. In our study, this community support was interestingly connected to race and cultural background. The SE'd workers from South Asian countries (e.g., Bangladesh, India) received substantial financial and emotional support from their community. For instance, as a disaster-prone country, Bangladesh is known for systems of informal support (e.g., kinship, community) in managing and resilience from natural (e.g. flood) and man-made (e.g. road accident) disasters [63,64]. Similarly, South Asian countries traditionally provide informal care to the elderly [46]. Although Canada is a high income and predominantly individualistic country, the large population of immigrant families from non-individualistic cultures may provide particular support for self-employment.

Finally, SE'd workers in this study relied heavily on their savings, which is consistent with previous studies showing that neoliberal mindsets of people encourage them to opt into SE and accept lesser reliance on government social security systems [10,33,65–69]. Neoliberal ideas encourage people to be self-dependent based on savings instead of relying on the states' contributions. This becomes particularly problematic when gig workers, often young people, become trapped in this unprotected form of work (e.g., gig workers) [3]. In this study and others, the so-called 'sustainability' of savings among SE'd workers was debunked, given the low and inconsistent income among some workers, including gig workers [16,67,70]. Indeed, several low-earning solo SE'd workers in our study vehemently expressed that their savings were insufficient to support the costs of prescription medicine and uninsured dental and other therapies. Hence, some were forced to depend on loans or credit cards to stay afloat, which, in turn, pushed them into the 'vicious cycle' of debt.

In all, while SE'd workers have never been a homogenous sector, there are now a greater portion of SE'd at the low-income end, such as in gig work. Such SE'd workers generally have no savings for rainy days, as their income is too low to allow for it. So, entrepreneurialism might be a difficult concept to apply to this new, growing low-end SE'd group. In the absence of social security coverage, this group is particularly reliant on informally developed support systems and have little access into supports for a decent life that are enjoyed by employees.

## Conclusion

To date, there has been little understanding of how solo SE'd workers experience and navigate income reduction or loss during periods of ill-health or injury. What support systems do SE'd workers seek and avail? Although scholars have engaged with the existing statuary or formal support systems for SE'd workers, little is known about their informal support systems. This study finds that, in the absence of adequate social security protections, SE'd workers relied substantially on informal support systems. While we cannot provide any single solution to better protect SE'd workers, our findings suggest that low-income workers could benefit from social security supports irrespective of employment status. Overall, SE'd workers, as a changing sector, requires consideration of equitable, inclusive, and sustainable social protection systems that ensure protection to meet people's needs over the life cycle.

## Acknowledgments

The authors thank to the participants of the study who shared their stories.

## Author Contributions

**Conceptualization:** Tauhid Hossain Khan.

**Data curation:** Tauhid Hossain Khan.

**Formal analysis:** Tauhid Hossain Khan.

**Funding acquisition:** Ellen MacEachen.

**Investigation:** Tauhid Hossain Khan.

**Methodology:** Tauhid Hossain Khan.

**Project administration:** Tauhid Hossain Khan.

**Resources:** Tauhid Hossain Khan.

**Software:** Tauhid Hossain Khan.

**Supervision:** Ellen MacEachen.

**Validation:** Tauhid Hossain Khan, Ellen MacEachen.

**Visualization:** Tauhid Hossain Khan.

**Writing – original draft:** Tauhid Hossain Khan.

**Writing – review & editing:** Tauhid Hossain Khan, Ellen MacEachen.

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
