## [Decision Letter · Decision Letter 0]

8 Sep 2023

PONE-D-23-16184The role of informal support systems during illness: a qualitative study of solo selfemployed

workers in Ontario, CanadaPLOS ONE

Dear Dr. Khan,

Thank you for submitting your manuscript to PLOS ONE. After careful consideration, we feel that it has merit but does not fully meet PLOS ONE’s publication criteria as it currently stands. Therefore, we invite you to submit a revised version of the manuscript that addresses the points raised during the review process. Please follow the comments raised by the referee before resubmitting.

We look forward to receiving your revised manuscript.

Kind regards,

Daphne Nicolitsas

Academic Editor

PLOS ONE

2. 1) In the ethics statement in the Methods, you have specified that verbal consent was obtained. Please provide additional details regarding how this consent was documented and witnessed, and state whether this was approved by the IRB

2) We noticed you have some minor occurrence of overlapping text with the following previous publication(s), which needs to be addressed:

- https://doi.org/10.1186/s12889-023-15471-8

- https://doi.org/10.3390/ijerph19095310

- http://dx.doi.org/10.1111/ecc.12746

In your revision ensure you cite all your sources (including your own works), and quote or rephrase any duplicated text outside the methods section. Further consideration is dependent on these concerns being addressed.

Reviewers' comments:

Reviewer's Responses to Questions

**Comments to the Author**

1. Is the manuscript technically sound, and do the data support the conclusions?

Reviewer #1: Yes

Reviewer #2: Yes

2. Has the statistical analysis been performed appropriately and rigorously? 

Reviewer #1: N/A

Reviewer #2: N/A

3. Have the authors made all data underlying the findings in their manuscript fully available?

Reviewer #1: Yes

Reviewer #2: Yes

4. Is the manuscript presented in an intelligible fashion and written in standard English?

Reviewer #1: Yes

Reviewer #2: Yes

5. Review Comments to the Author

Reviewer #1: The manuscript investigated the informal support systems used by self-employed individuals during illness/injury and reduction or loss of income. The manuscript provides an overview of existing literature, and the methodology is clearly described, justified and appropriate for the research problem. Findings are presented in a structured way, supported by verbatim from the interviews and is in line with the standard for reporting qualitative research. Conclusions are supported by and aligned with the findings. The findings and conclusions contribute to an important, but often overlooked aspect of entrepreneurship/SE.

The following minor corrections may be considered:

- change format of brackets for citations from ( ) to [ ]

- The following lines include the author surnames and date in the citation, instead of only the reference list number:

Lines 83, 84, 114, 328, 332, 333, 335, 336, 338, 341, 385

- Line 91: Hilbrecht (2016) [15] - also include the reference list number after the date. The same applies to

Line 363, Akram and Maitrot (2022) [58]

- Line 95: ...towards SE'd workers - remove 'a'

- Line 104: ...(36). (37). - both sources [36,37]

- Line 146: ...three key themes.../...three major themes... - Choose either key or major, not both.

- Line 358: Kim, Longest and Aldrich [47]... - Aldrich's surname was left out.

- Line 391: ...'vicious cycle' of loans. - Consider 'vicious cycle' of debt (instead of loans)

Reviewer #2: This is a very interesting study which addresses the importance of health related issues for self-employed workers. The study clearly demonstrates the importance of governmental social welfare programs as well as informal social networks for providing health care. One of the issues that I think is crucial is the fact that outside of governmental support it is very difficult to provide health care that is necessary to meet the needs of all individuals in society. I strongly agree with the authors that social networks that individuals build are indeed important for any person's recovery during health-related issues; however, this is not sufficient for meeting other more complex health problems that can only be addressed by government funded health care systems if a society is serious about keep all individuals healthy. A case and point is the recent pandemic where many governments around the world funded the vaccination of millions of people to prevent massive loss of lives. Without such funding the toll of deaths resulting from the pandemic would have been much greater.

6. PLOS authors have the option to publish the peer review history of their article (what does this mean?). If published, this will include your full peer review and any attached files.

Reviewer #1: No

Reviewer #2: No

---

## [Decision Letter · Decision Letter 1]

12 Jan 2024

The role of informal support systems during illness: A qualitative study of solo self-employed

workers in Ontario, Canada

PONE-D-23-16184R1

Dear Dr. Khan,

We’re pleased to inform you that your manuscript has been judged scientifically suitable for publication and will be formally accepted for publication once it meets all outstanding technical requirements.

Kind regards,

Daphne Nicolitsas

Academic Editor

PLOS ONE

Additional Editor Comments (optional):

Reviewers' comments:

Reviewer's Responses to Questions

**Comments to the Author**

1. If the authors have adequately addressed your comments raised in a previous round of review and you feel that this manuscript is now acceptable for publication, you may indicate that here to bypass the “Comments to the Author” section, enter your conflict of interest statement in the “Confidential to Editor” section, and submit your "Accept" recommendation.

Reviewer #1: All comments have been addressed

Reviewer #3: All comments have been addressed

2. Is the manuscript technically sound, and do the data support the conclusions?

Reviewer #1: Yes

Reviewer #3: Yes

3. Has the statistical analysis been performed appropriately and rigorously? 

Reviewer #1: N/A

Reviewer #3: Yes

4. Have the authors made all data underlying the findings in their manuscript fully available?

Reviewer #1: Yes

Reviewer #3: Yes

5. Is the manuscript presented in an intelligible fashion and written in standard English?

Reviewer #1: Yes

Reviewer #3: Yes

6. Review Comments to the Author

Reviewer #1: I am satisfied that all previous comments have been addressed.

However, the following sources are not included in the References:

p4, line 64 - (LCO, 2012:75)

p7, line 114 - (Patton, 2001)

The following inconsistencies with citations:

p18, line 360 - PH Kim, KC Longest and HE Aldriich [52] - initials and reference number included

p18, line 362 - M Hilbrecht [16] - initials and reference number included

p18, line 366 - Akram and Maitrot (2022) - no intials and no reference number

The following typos can be corrected:

p12, line 207 - ...likecancer... - needs a space in between 'like' and 'cancer'

p18, line 355 - ...suggested that that some... - remove extra 'that'

Reviewer #3: REVIEWER COMMENTS ON MANUSCRIPT ENTITLED “THE ROLE OF INFORMAL SUPPORT SYSTEMS DURING ILLNESS: A QUALITATIVE STUDY OF SOLO SELF-EMPLOYED WORKERS IN ONTARIO, CANADA”

Strengths and weakness of the paper

Thanks for the opportunity to read this manuscript of yours. The study was on a very important topic that appeals to an international readership. Very important data was collected for the study and the results presented were very important for policy and practical implications. I therefore make the following suggestions below to improve the paper and bring it to the publishing level.

Abstract

1. The philosophical position, and research approach, used should be indicated in the abstract.

Background

2. Check from the journal style if the subheading should not be introduction instead of background.

3. The background should end with a clear policy gap in the study area that this study seeks to fill. This is because the actual gap that this study seeks to fill is missing in the study.

4. Clear specific research questions or objectives could be very helpful.

5. The greatest challenge for this paper is the lack of a literature review section on the key variables of the study.

6. A special section on the review of social policy in Canada's social security system could be helpful.

Methodology and data

7. The specific research design should be indicated in the methods section of the paper.

8. It is not clear how the participants were selected in the methodology section.

9. What sampling technique/s was/were used for the study and it was not clear how the 24 participants were arrived at.

10. The sections in the research instrument indicated on page 8 should have guided the framing of research questions or objectives to guide the study.

11. It was also not clear how validity and reliability were achieved for this study.

Results and discussion

12. The results presented were very important. I only think that if there were specific research questions guiding the study, they could have improved the results sections.

THANKS, AND HOPE THESE COMMENTS HELP YOU BRING THE PAPER TO A PUBLISHABLE STATE.

7. PLOS authors have the option to publish the peer review history of their article (what does this mean?). If published, this will include your full peer review and any attached files.

Reviewer #1: No

Reviewer #3: **Yes: **Moses Segbenya

---

## [Editor Report · Acceptance letter]

4 Mar 2024

PONE-D-23-16184R1 

PLOS ONE

Dear Dr. Khan, 

I'm pleased to inform you that your manuscript has been deemed suitable for publication in PLOS ONE. Congratulations! Your manuscript is now being handed over to our production team.

Kind regards, 

on behalf of

Dr. Daphne Nicolitsas 

Academic Editor

PLOS ONE